# Urinary Titin N-Fragment as a Biomarker of Muscle Atrophy, Intensive Care Unit-Acquired Weakness, and Possible Application for Post-Intensive Care Syndrome

**DOI:** 10.3390/jcm10040614

**Published:** 2021-02-06

**Authors:** Nobuto Nakanishi, Rie Tsutsumi, Kanako Hara, Masafumi Matsuo, Hiroshi Sakaue, Jun Oto

**Affiliations:** 1Emergency and Critical Care Medicine, Tokushima University Hospital, 2-50-1 Kuramoto, Tokushima 770-8503, Japan; joto@tokushima-u.ac.jp; 2Department of Nutrition and Metabolism, Tokushima University Graduate School of Biomedical Sciences, 3-18-15 Kuramoto, Tokushima 770-8503, Japan; rtsutsumi@tokushima-u.ac.jp (R.T.); c201931015@tokushima-u.ac.jp (K.H.); hsakaue@tokushima-u.ac.jp (H.S.); 3Research Center for Locomotion Biology, Kobe Gakuin University, 518 Arise, Ikawadani, Nishi, Kobe 651-2180, Japan; mmatsuo@reha.kobegakuin.ac.jp

**Keywords:** titin, muscle, diaphragm, atrophy, physical dysfunction, biomarker, urine, post-intensive care syndrome, nutrition, rehabilitation

## Abstract

Titin is a giant protein that functions as a molecular spring in sarcomeres. Titin interconnects the contraction of actin-containing thin filaments and myosin-containing thick filaments. Titin breaks down to form urinary titin N-fragments, which are measurable in urine. Urinary titin N-fragment was originally reported to be a useful biomarker in the diagnosis of muscle dystrophy. Recently, the urinary titin N-fragment has been increasingly gaining attention as a novel biomarker of muscle atrophy and intensive care unit-acquired weakness in critically ill patients, in whom titin loss is a possible pathophysiology. Furthermore, several studies have reported that the urinary titin N-fragment also reflected muscle atrophy and weakness in patients with chronic illnesses. It may be used to predict the risk of post-intensive care syndrome or to monitor patients’ condition after hospital discharge for better nutritional and rehabilitation management. We provide several tips on the use of this promising biomarker in post-intensive care syndrome.

## 1. Introduction

Titin, also called connectin, is a giant sarcomere protein, which functions as a spring for muscle extension and elasticity [1]. Titin interconnects the contraction of actin-containing thin filaments and myosin-containing thick filaments. Recently, the N-terminal fragment of titin, which is the breakdown product of titin, has become measurable using an enzyme-linked immunosorbent assay kit (27900 titin N-fragment Assay Kit; Immuno-Biological Laboratories, Fujioka, Japan) [2]. This kit has been used to evaluate muscle breakdown in muscle dystrophy, in which the level of urinary titin N-fragment was 700-times above the normal level [3].

Muscle atrophy and weakness are common in critically ill patients [4,5,6]. In particular, muscle weakness has been widely recognized as an intensive care unit-acquired weakness (ICU-AW) [7]. Although the pathophysiology of ICU-AW is still unknown, Swist et al. found titin loss in the muscle biopsies of critically ill patients, and suggested that the titin loss was a cause of ICU-AW [8]. Two study groups have recently reported the use of urinary titin N-fragment in assessing ICU-AW. Nakanishi et al. found that the urinary titin N-fragment reflected muscle atrophy and ICU-AW in critically ill patients [9]. Moreover, the accumulated urinary titin was associated with mortality in these patients. Another study group, Nakano et al., reported that urinary titin could be a possible biomarker of muscle atrophy and ICU-AW [10,11]. Urinary titin N-fragment may become an important test in the ICU.

The ICU-AW remains years after ICU discharge, and is known as post-intensive care syndrome (PICS) [12]. PICS is characterized by physical dysfunctions, psychological disorders, or cognitive impairments that persist beyond ICU discharge. The state persists, even five years after ICU discharge [13]. One of the important measures for reducing PICS is to follow up high-risk patients after ICU discharge. The urinary titin N-fragment can be used as a biomarker to identify patients who exhibit increased catabolism and need intervention. We suggest several tips on how to efficiently use the urinary titin N-fragment in PICS. In this review, we have summarized recent literature on titin, and the possible applications of the urinary titin N-fragment in PICS.

## 2. Titin

Titin, initially known as connectin, was discovered in 1976 by Maruyama et al. [14]. Being the largest protein in the human body, it was renamed titin after the giant god, Titan, from Greek mythology. Titin is the largest protein in humans, and is 3.0–3.7 MDa. This protein is located in the muscle sarcomere, and interconnects the contraction of actin-containing thin filaments and myosin-containing thick filaments. Passive tension and elasticity during muscle contraction develops as a result of the titin protein by Ca^2+^-dependent stiffening (Figure 1) [1,15].

During muscle degradation, titin is broken down into small fragments, and several different fragments are measurable. In serum, the metalloproteinase (MMP) 2-cleaved titin fragment and MMP 12-cleaved titin fragment are measurable. MMP 2-cleaved titin reflected muscle atrophy in a human bed rest study [16]. On the other hand, MMP 12-cleaved titin fragment could be used to assess cardiac infarction, because the level of the fragment increased after an acute myocardial infarction [17].

Recently, the N-terminal fragment of titin, which is 25 kDa, and is known as the urinary titin N-fragment, has become measurable in urine. Unlike serum, this urinary biomarker is noninvasive, but requires correction by urinary creatinine to adjust for the kidney function. The standard value of urinary titin N-fragment was 2.1 (1.2–2.6) pmol/mg Cr in healthy adult volunteers [2].

## 3. Muscle Atrophy

### 3.1. Limb and Trunk Muscle Atrophy

Muscle atrophy is a serious problem in critically ill patients [4]. After one week of ICU admission, critically ill patients exhibited 13.2–16.9% of muscle atrophy in the upper limbs, and 18.8–20.7% in the lower limbs [18]. These muscle atrophies are associated with impaired physical function and mortality [19,20].

Muscle atrophy is caused by an increase in protein degradation or decrease in protein synthesis. Increased protein degradation occurs mainly due to inflammation and immobilization. Calpain, caspase, ubiquitin-proteasome, and autophagy-lysosome have been implicated in this protein degradation pathway [21,22,23]. Decreased protein synthesis is caused by suppressed insulin-like growth factor-1 and inactivated myogenesis [24].

In critically ill patients, inflammation is an important cause of muscle atrophy [25], and various inflammatory cytokines are associated with muscle atrophy [26]. Immobilization is frequently observed during critical illness, and causes muscle atrophy [27]. Malnutrition causes a decrease in protein synthesis [28]. In critically ill patients, recommended protein intake is 1.2–2.0 g/kg/day [29], but this level of intake is often not achieved in the ICU [30], resulting in decreased protein synthesis.

Muscle mass can be assessed using computed tomography, bioelectrical impedance analysis, and ultrasound [31]. Computed tomography is accurate, but exposes patients to radiation, whereas bioelectrical impedance analysis is influenced by fluid changes in critically ill patients [32]. Ultrasound can be used to monitor muscle atrophy at the bedside, but it requires a skilled and experienced operator [33,34]. Thus, a biomarker to assess muscle atrophy is urgently needed. In a rat study, Udaka et al. found that six weeks of immobilization caused titin loss in the soleus muscle, which was associated with muscle dysfunction [35]. Thus, it is theoretically reasonable to expect levels of urinary titin to be elevated in the urine of patients with muscle atrophy.

In muscular dystrophy, the urinary titin N-fragment reflects the disease severity. Patients with Duchenne muscular dystrophy had a higher concentration of the urinary titin N-fragment than those with Becker muscular dystrophy (965.8 vs. 171.2 pmol/mg Cr, *p* < 0.01) [36]. This result possibly reflects the amount of muscle breakdown in these conditions. Recently, two studies have reported the usefulness of the urinary titin N-fragment in the evaluation of muscle atrophy in critically ill patients. Furthermore, Nakano et al. reported that urinary titin N-fragment could be used to evaluate muscle atrophy in critically ill patients [11]. In their study, investigating four critically ill patients, there was a negative correlation between mean urinary titin level during the first seven days of ICU admission and femoral muscle volume measured using computed tomography (*r* = −0.729). Furthermore, Nakanishi et al. reported that in 56 nonsurgical critically ill patients, the cumulative urinary titin concentration on days 3, 5, and 7 was significantly higher in the prominent muscle atrophy group (*p* ≤ 0.03), suggesting that urinary titin reflects muscle atrophy in nonsurgical critically ill patients [9]. However, in their study, the correlation between muscle atrophy and urinary titin was limited to *r* = 0.29–0.54 (*p* ≤ 0.03), suggesting that urinary titin levels are affected by various physiologic conditions. As inflammation is an important cause of muscle atrophy, the peak urinary titin N-fragment level was higher in patients with sepsis (93.0 vs. 57.9 pmol/mg Cr, *p* = 0.02). Moreover, the high levels of urinary titin N-fragment were associated with increased mortality. Although further studies are required, it is clear that a relationship exists between muscle atrophy and urinary titin N-fragment.

The molecular mechanism underlying titin breakdown remains unclear. Several pathways, activated by inflammation and immobilization, are involved in the breakdown of titin. Calpain contributes to the cleavage of titin because titin has calpain-binding sites [37]. Lang et al. investigated the ubiquitination of titin in denervated mouse, and found that levels of ubiquitinated titin gradually increased in denervation-induced muscle atrophy [38]. Consistent with this finding, Swist et al. found increased levels of ubiquitinated titin in patients with critical illness [8]. In their study, the markers of the autophagy-lysosome pathway were also upregulated. Thus, the autophagy-lysosome pathway may be involved in the breakdown of titin.

Unlike other promising biomarkers of muscle atrophy, urinary titin N-fragment is noninvasive and reliable. Creatinine kinase and BUN/Cr are possible biomarkers for muscle atrophy [39], but these biomarkers require blood tests. Furthermore, creatinine kinase is derived from various tissues [40], and BUN/Cr is influenced by various conditions including dehydration [39]. Urinary creatinine has also been suggested to be a biomarker of muscle atrophy, but it does not consider kidney function [41]. Thus, urinary titin N-fragment, corrected by urinary creatinine, is a reliable biomarker because it does not depend on kidney function [9,10].

### 3.2. Diaphragm Muscle Atrophy

Diaphragm atrophy is observed in 60% of mechanically ventilated critically ill patients [42], and it is a serious problem because of its association with prolonged mechanical ventilation and prolonged ICU stay [43]. As with limb muscle atrophy, diaphragm atrophy is caused by the calpain, caspase, ubiquitin-proteasome, and autophagy-lysosome pathways [44,45,46,47]. As reported in limb muscles, inflammation and immobilization are important causes of diaphragm atrophy. Sepsis is a cause of diaphragm atrophy [48], and deep sedation causes immobilization of the diaphragm [49]. Most importantly, ventilator settings strongly influence diaphragm atrophy and subsequent diaphragm dysfunction. Thus, diaphragm dysfunction in such cases is termed as ventilator induced diaphragm dysfunction [50].

Titin plays an important role in the diaphragm contractile force [51], and titin loss has been associated with diaphragm dysfunction in rats [52,53]. In the diaphragm biopsy of human subjects, Hussain et al. found that prolonged controlled mechanical ventilation decreased titin levels and impaired the diaphragm myofibrillar force [54]. Furthermore, Lindqvist et al. suggested that the positive-end expiratory pressure (PEEP) during ventilation led to the breakdown of the diaphragm titin, because the PEEP stretched out the sarcomere of the diaphragm muscle fibers [55]. Excessive extension may be detrimental to diaphragm titin.

Although titin is associated with diaphragm atrophy and dysfunction, urinary titin N-fragment is not useful for detecting diaphragm atrophy. Our previous study investigated the change of diaphragm thickness in 50 critically ill patients using ultrasound. Diaphragm atrophy, defined by a >10% decrease of diaphragm thickness, was observed in 32 patients (64%), and the mean diaphragm thickness decreased by 4.9% ± 15.8%, 8.0% ± 16.9%, and 15.4% ± 10.2% on days 3, 5, and 7, respectively. On comparing the diaphragm atrophy and unchanged groups, the levels of urinary titin N-fragment were not higher in the diaphragm atrophy group than those in the unchanged group (147.9 vs. 192.4 pmol/mg in the unchanged vs. atrophy group, *p* = 0.33) [9]. The urinary titin N-fragment can measure the titin breakdown product in all muscles, and is not specific to the diaphragm. To quantify the diaphragm atrophy, a diaphragm-specific titin kit is necessary. This may be theoretically possible, because a cardiac-specific titin kit has been developed in another study [56].

Interestingly, several studies have reported that, as well as diaphragm atrophy, increased diaphragm thickness has worsened clinical outcomes [42,43,57]. Insufficient ventilatory support leads to excessive respiratory effort in mechanically ventilated patients. This condition increases the diaphragm thickness. Since the increased muscle thickness has worsened outcomes, the hypertrophied diaphragm may not have sufficient functional titin to function appropriately. In a previous study on urinary titin N-fragment, there was no significant difference in the cumulative urinary titin N-fragment between the unchanged diaphragm thickness and increased diaphragm thickness groups (147.9 (79.0–257.8) vs. 426.1 (140.8–578.2) pmol/mg Cr in unchanged vs. increased, *p* = 0.45) [9]. The combination of increased diaphragm thickness and atrophy also did not have a significant difference in terms of the cumulative level of urinary titin N-fragment (147.9 (79.0–257.8) vs. 206.5 (99.3–440.8) pmol/mg Cr in unchanged vs. combination, *p* = 0.31). The mechanism underlying the increase in diaphragm thickness remains to be elucidated.

Diaphragm dysfunction is preventable and reversible. Therefore, it is important to maintain spontaneous breathing and avoid excessive ventilatory support during mechanical ventilation, which is called diaphragm protective ventilation [43]. Diaphragm protective ventilation can prevent diaphragm atrophy, compared with lung protective ventilation [58]. Furthermore, O’ Rourke et al. reported that percutaneous electrical phrenic nerve stimulation increased diaphragm thickness by 15.1% within 48 h [59]. Extracorporeal support is also considered to prevent diaphragm injury [60], and in a case report, the early initiation of extracorporeal support prevented diaphragm atrophy, with a relatively suppressed level of urinary titin N-fragment of 24.1–38.4 pmol/mg Cr [61].

### 3.3. Other Respiratory Muscle Atrophy

In addition to the diaphragm muscle, intercostal muscle atrophy is also observed in patients with mechanical ventilation [62], and is associated with prolonged mechanical ventilation and prolonged ICU stay [42]. Moreover, muscle atrophy occurs in other expiratory muscles, including the obliquus interna, obliquus externa, transversus abdominis, and rectus abdominis muscles [63]. In the case report of a mechanically ventilated patient, intercostal muscle biopsy showed the loss of myosin-containing thick filaments, with the possible detachment of titin [64]. Titin loss may be an important cause of other respiratory muscle dysfunctions, as well as that of the diaphragm. Jonkman et al. reported that breath-synchronized electrical stimulation increased the thickness of abdominal expiratory muscles (1.76 mm vs. −0.50 mm in intervention vs. control, respectively, *p* = 0.02) [65]. Thus, titin loss may be reversible by active rehabilitation.

## 4. ICU-Acquired Weakness

In the ICU, newly acquired muscle weakness is called ICU-AW, which is found in 40–50% of all critically ill patients [4]. In a previous study, ICU-AW was independently associated with physical dysfunction at six months after ICU discharge [66]. The diagnosis of ICU-AW requires muscle strength assessment using a medical research council score <48 [67], and a low medical research council score is associated with impaired physical functions, including handgrip strength, 6-min walking distance, and physical functioning of SF-36, even five years after ICU discharge [68]. Moreover, because the diagnosis of ICU-AW requires the active cooperation of patients, the assessment is not feasible in approximately half of the critically ill patients [4]. Thus, the development of a biomarker is important to diagnose ICU-AW. However, there has been no established available biomarker to diagnose ICU-AW [69].

ICU-AW is classified into critical illness myopathy or neuropathy [7]. Although the underlying mechanism is still unknown, damage to myosin-containing thick filaments has been proposed to contribute to critical illness myopathy [70,71]. Recently, Swist et al. collected biopsies of the tibialis anterior muscles in nine mechanically ventilated ICU patients diagnosed with critical illness myopathy, and found that not only the levels of myosin-containing thick filaments, but also those of titin, were lost in the muscle, whereas the levels of actin-containing thin filaments were unchanged [8]. In their other study, a titin-inactivated mouse had sarcomere disintegration, myocyte de-stiffening, and force impairment, which were consistent with the muscle damage of critical illness myopathy. Thus, they suggested that titin loss is a contributing factor to critical illness myopathy. With regard to critical illness neuropathy, Chen et al. investigated the role of titin in neuropathy [72]. They denervated the tibialis anterior muscles of rats, and measured the amount of titin in the muscle. In the denervated muscle, titin loss was observed, and the loss was dependent on the denervation time. They found that titin was translocated and possibly cleaved from the sarcomere, and concluded that titin was sensitive to degradation after denervation.

Urinary titin N-fragment is useful to assess functional impairments. Ishihara et al. reported that the level of urinary titin N-fragment correlated with functional impairments in patients after stroke [73]. In the study, peak urinary titin N-terminal fragment levels during the seven days of admission were correlated with modified Rankin scale score (*r* = 0.55, *p* < 0.01), National Institute of Health stroke scale score (*r* = 0.72, *p* < 0.01), and Barthel index (*r* = −0.59, *p* < 0.01) at the time of hospital discharge. In the multivariate analysis adjusted for the disease severity, the urinary N-terminal fragment on day 2 predicted the functional outcome at hospital discharge (odds ratio, 1.11; 95% CI, 1.01–1.28). This study excluded patients with: in-hospital onset, dialysis, surgery, and seizure, as well as those not having independent daily living. Thus, it is reasonable to believe that the urinary titin N-fragment level reflected muscle breakdown and subsequent functional impairments.

Two study groups have recently reported the usefulness of urinary titin N-fragment levels in ICU-AW. Nakanishi et al. investigated the urinary titin N-fragment in 56 nonsurgical critically ill patients, and found that the cumulative urinary titin N-fragment up to discharge or day 7 was higher in ICU-AW than non-ICU-AW patients (314.1 (181.5–464.7) vs 86.6 (66.3–171.1), *p* = 0.01) [9]. In the study, urinary titin level on day 2 predicted ICU-AW with a sensitivity of 78% and specificity of 81% at the cut-off value of 64.8 pmol/mg Cr. In another study, Nakano et al. investigated 50 consecutive critically ill patients, and found that the medical research council score was lower in the high urinary titin N-fragment group (37.0 (24.0–56.0) vs. 56.0 (51.0–60.0), *p* = 0.023) [10]. In multivariate analysis, urinary titin N-fragment was independently associated with medical research council score <48, almost equivalently with ICU-AW, after adjusting for age, sex, sequential organ failure assessment score, and steroid dose (adjusted odds ratio: 1.02 (95% CI: 1.00–1.03), *p* = 0.02). In their study, the mean urinary titin N-fragment level during the first 7 days of ICU admission predicted ICU-AW with a sensitivity of 61.9% and specificity of 89.7% at the cut-off value of 100 pmol/mg Cr. The area under the curve to predict medical research council score <48 was 0.810 (95% CI: 0.688–0.931), whereas creatine kinase had an area under the curve of 0.654 (95% CI: 0.494–0.814). As titin is a functional protein, it is theoretically understandable that urinary titin is a better biomarker of ICU-AW than creatine kinase, which is an enzyme found in various tissues [40]. Indeed, another study reported that creatinine kinase level did not differ in ICU-AW or non-ICU-AW patients (405 vs. 508 pmol/mg Cr, *p* = 0.10) [74].

It would be beneficial if we can predict ICU-AW using the urinary titin N-fragment because ICU-AW is a preventable condition. Nutritional support and rehabilitation can prevent ICU-AW [75]. In patients at risk of ICU-AW, we can provide intense nutrition and rehabilitation management. Moreover, we can avoid, or at least reduce, the risk medications including catecholamines, steroids, or neuromuscular blockers [71,76,77]. The measurement of urinary titin N-fragment may change our clinical practice regarding the management of critically ill patients.

## 5. PICS

PICS is characterized by physical, psychological, or cognitive impairments persisting beyond ICU discharge [12]. Survivors of acute respiratory distress syndrome have prolonged muscle atrophy, impaired gait speed, and a deteriorated 6-min walk distance at 6–12 months after ICU discharge [78]. These physical dysfunctions can persist for up to five years in certain patients [13]. In a study, 46% of patients had persistent symptoms of ICU-AW for 5–10 years [79]. These prolonged physical dysfunctions hinder the ability of patients to return to work [80].

Recently, PICS has been gaining increased attention because of the aging society and the coronavirus disease 2019 (COVID-19) pandemic [81]. The elderly population is increasing worldwide, and this population has a high risk of PICS [12]. At 12 months after hospital discharge, the Barthel index was lower in patients aged >75 years than in those aged 65–74 years (*p* < 0.01) [82]. In COVID-19, muscle weakness <48 was observed in 66% of patients [83], and ICU-AW was observed in 52% of patients at ICU discharge and 27% of patients at hospital discharge after COVID-19 infection [84]. After the COVID-19 pandemic, a PICS pandemic is anticipated, requiring some preparation [85].

The loss of titin may be a cause of the prolonged physical dysfunction of PICS. We considered two hypotheses: (1) prominent muscle breakdown in the ICU, (2) prolonged muscle breakdown after hospital discharge (Figure 2).

First, prominent muscle breakdown will lead to increased muscle atrophy and weakness. The change in the acute phase may persist into PICS. There are several strategies to prevent the prominent muscle breakdown in the acute phase. Rehabilitation has been proven to be effective in preventing muscle atrophy and physical dysfunction [86,87]. Nakanishi et al. reported that electrical muscle stimulation was effective in preventing muscle breakdown [88]. In the study, blood branched-chain amino acid, which is an important muscle component, was investigated, and the amino acid level was lower in the patients who received the electrical muscle stimulation intervention (40.5% (−7.4–75.3%) vs. 71.5% (38.8–116.9%)), suggesting a decrease in muscle breakdown. Another strategy is to ensure sufficient protein intake during the acute phase. Nakamura et al. reported that protein intake of 1.5 g/kg/day prevented muscle atrophy during the first 10 days of ICU admission, compared with 0.8 g/kg/day (12.9% ± 8.5% vs. 16.9% ± 7.0% in 1.5 vs. 0.8 g/kg/day, *p* < 0.01) [89]. These strategies in the acute phase will possibly prevent PICS.

Second, patients with PICS may have prolonged muscle breakdown due to the chronic inflammation or decreased mobility. Some patients experience chronic inflammation, which is also known as persistent inflammation, immunosuppression, and catabolism syndrome [90]. This condition may be caused by various conditions including disseminated intravascular coagulopathy [91] and electrolyte imbalance [92]. Chronic inflammation is known to cause muscle atrophy [93]. Decreased mobility is also a cause of prolonged muscle breakdown. After hospital discharge, impaired physical function or endurable pain will lead to decreased mobility [94]. Although early mobilization is important to prevent physical dysfunction, Fuke et al. reported that early mobilization is not sufficient to prevent PICS [95]. One of the strategies to prevent PICS is following up the patient after the hospital discharge. Although PICS follow-up has been shown to decrease mortality and medical cost [96], few facilities conduct PICS follow-up. In most facilities, human resources and financial support are not sufficient to conduct PICS follow-up [97]. Thus, the international consensus conference of critical care medicine recommended selecting the patients to follow up, and that the selection should be conducted within 2–4 weeks after hospital discharge [98].

We consider that the detection of the level of urinary titin N-fragment may be important for the screening, follow-up, and management. Urinary titin N-fragment reflects ongoing muscle breakdown because the fragment can detect muscle breakdown within at least 2 h of its onset [73]. The measurement of urinary titin N-fragment can identify patients at high risk of prolonged muscle breakdown. In a previous report, Oshida et al. investigated the level of urinary titin N-fragment in patients with non-alcoholic fatty liver disease [99]. They found that the urinary titin N-fragment was negatively correlated with skeletal muscle mass (*r* = −0.134, *p* < 0.05), grip strength (*r* = −0.203, *p* < 0.01), and knee extension muscle strength (*r* = −0.191, *p* < 0.05), and positively correlated with the echo intensity of the rectus femoris muscle (*r* = 0.361, *p* < 0.001), which indicates the fibrous changes of the muscle tissue [100]. Although the reported average of urinary titin N-fragment concentration was 2.1 (1.2–2.6) pmol/mg/Cr, the level was increased several times in some patients with, and without, non-alcoholic fatty liver disease. Another study by Miyoshi et al. investigated patients with gastrointestinal tract and hepatobiliary pancreatic malignancies, and found that the urinary titin N-fragment was significantly higher in sarcopenia (8.3 (1.9–20) vs. 4.9 (2.3–15) pmol/mg Cr, *p* = 0.04) [101]. In their study, urinary titin N-fragment showed statistically significant negative correlations with skeletal muscle volume index (*r* = −0.16, *p* = 0.04). These two studies suggest that the urinary titin N-fragment may be utilized to assess the muscle breakdown state in PICS.

Before we use the urinary titin N-fragment in clinical practice, we need to know several factors. First, a spot urine test is available for urinary titin N-fragment. Although the 24-h urine sample appears reliable, it is not feasible during follow-up. In urinary titin N-fragment, circadian variations are limited [102]. Second, surgical procedures may increase the urinary titin N-fragment. This is theoretically reasonable, and urinary titin N-fragment was indeed elevated shortly after cardiac surgery [103]. Third, the standard level of the urinary titin N-fragment differs according to age. Age is associated with muscle atrophy due to increased catabolism [104]. Indeed, in a previous study, the average urinary titin N-fragment level was 2.3 ± 1.4 pmol/mg Cr in patients aged <30 years old, 4.3 ± 3.7 pmol/mg Cr in those aged 31–60 years, and 5.7 ± 4.0 pmol/mg Cr in those aged >60 years [99]. In another study, urinary titin N-fragment was correlated with age (*r* = 0.11, *p* = 0.04) [101]. Fourth, active exercise should be refrained from before the measurement of urinary titin N-fragment, because exercise increases the urinary titin N-fragment [105,106]. Due to the elastic property of titin, eccentric exercises are more injurious to the titin of muscles than concentric exercise [107]. Another study investigated the change of urinary titin N-fragment in people playing a soccer match, and they found that the urinary titin N-fragment was increased during the 24 h after the soccer match, and returned to the baseline value at 48 h after the match [108]. The increased urinary titin N-fragment generally remains for 2–3 days after exercise [2]. Fifth, urinary titin N-fragment is also increased by cardiac damage, because the urinary titin N-fragment reflects the breakdown products of titin, including the cardiac source. Titin loss is a contributing factor of dilated cardiomyopathy [109]. In a previous study, increased urinary titin N-fragment predicted the mortality in patients with dilated cardiomyopathy (*p* < 0.05) [110]. We summarized several factors that lead to an increase in the urinary titin N-fragment, as supported by various studies (Table 1).

We can intervene if patients have increased urinary titin N-fragment after hospital discharge. Nutrition and rehabilitation interventions are important in PICS [111]. After hospital discharge, continuous physical therapy is important in the home or community-based settings [112]. In the exploratory analyses of a randomized controlled trial, continuous rehabilitation after hospital discharge led to an improvement of the 6-min walking distance at the 12 month follow-up [113]. Some patients may have persisting inflammation, but exercise has anti-inflammatory effects [114], and physical therapy is expected to reduce the inflammatory reaction [115]. Nutritional support prevents muscle breakdown in the chronic phase [116]. Regarding titin, Ulanova et al. reported that L-arginine administration decreased the loss of titin in rat soleus muscle [117]. Since nutritional support team involvement can improve calorie and protein delivery [118,119], a multidisciplinary team intervention is expected to be important in PICS if urinary titin is elevated in patients.

## 6. Conclusions

Titin plays an important role in the muscles, and urinary titin N-fragment can measure muscle breakdown in critically ill patients, and possibly in PICS. As reported in muscle dystrophy, urinary titin N-fragment reflected the extent of muscle atrophy in critically ill patients. Furthermore, urinary titin N-fragment reflected the muscle weakness, consistent with the functional role of the urinary titin N-fragment. It is important to note that the urinary titin N-fragment increases with age, exercise, surgery, and cardiac damage, in addition to muscle atrophy and weakness. Our results indicate that urinary titin N-fragment may be a marker in PICS for identifying patients with increased catabolism, and requiring interventional support.

## Figures and Tables

**Figure 1 jcm-10-00614-f001:**
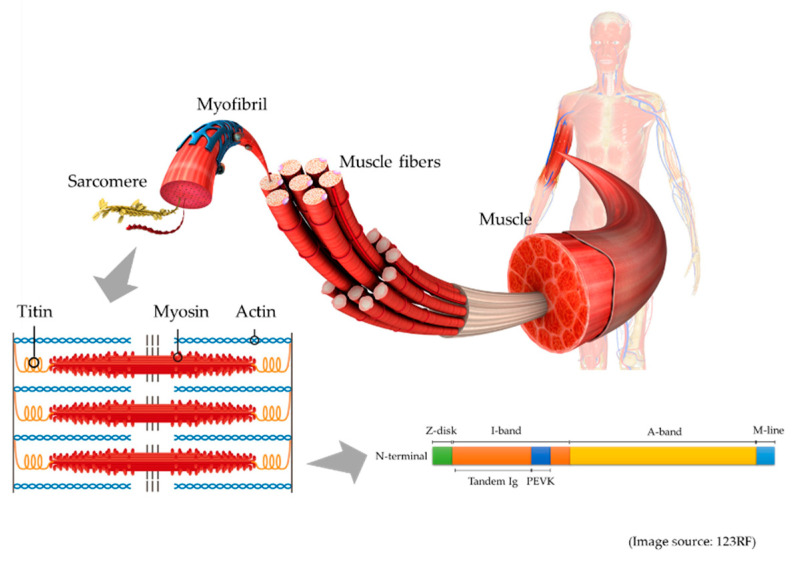
This is a schematic illustration of titin in the muscle. Muscle comprises of muscle fibers, myofibrils, and the smallest units, known as sarcomeres. In the sarcomere (lower left), titin connects actin-containing thin filaments and myosin-containing thick filaments. In the schematic illustration of titin structure (lower right), titin is composed of Z-disk, I-band, A-band, and M-line regions, and the I-band includes tandem Ig and PEVK (Pro-Glu-Val-Lys) domains.

**Figure 2 jcm-10-00614-f002:**
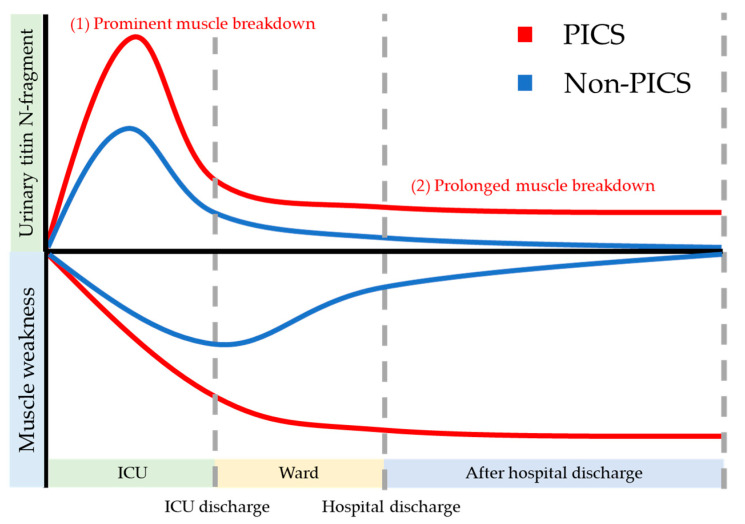
This is a conceptual image of urinary titin N-fragment and muscle weakness in PICS. In PICS, urinary titin N-fragment may be elevated due to the prominent muscle breakdown in the acute phase or prolonged muscle breakdown in the chronic phase, leading to the muscle weakness after hospital discharge. PICS: post-intensive care syndrome.

**Table 1 jcm-10-00614-t001:** Factors leading to an increase in the urinary titin N-fragment.

Factors	Level of Urinary Titin N-Fragment	Evidence
Age	2.3, 4.3, 5.7 pmol/mg/Cr in ≤30, 31–60, ≥61 years old	[99]
Exercise	40–100 pmol/mg/Cr in exercise-induced muscle damage	[106,107]
Surgery	30–50 pmol/mg/Cr after cardiac surgery	[103]
Cardiac damage	≥7.26 pmol/mg/Cr in a third of dilated cardiomyopathy	[110]

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
