# Peer review of "Urinary Titin N-Fragment as a Biomarker of Muscle Atrophy, Intensive Care Unit-Acquired Weakness, and Possible Application for Post-Intensive Care Syndrome"

_jcm, 2021, doi:10.3390/jcm10040614_

Round 1

Reviewer 1 Report

It is an interesting paper regarding the role of titin urine metabolite in the evaluation of patients hospitalised in the ICU. N-titin is a marker of mascular metabolism/catabolism.

The clinical significance of the chosen review of literature and the presentation of the subject is thorough. Overall, I believe it adds to the literature and the manuscript should be accepted for publication by your esteemed journal.

Author Response

Responses to Reviewer #1:

  1. It is an interesting paper regarding the role of titin urine metabolite in the evaluation of patients hospitalised in the ICU. N-titin is a marker of mascular metabolism/catabolism. The clinical significance of the chosen review of literature and the presentation of the subject is thorough. Overall, I believe it adds to the literature and the manuscript should be accepted for publication by your esteemed journal.
  2. We appreciate reviewer’s positive comments.

Reviewer 2 Report

  1. The manuscript requires extensive editing. Many of the sentences are unclear and the expression "on the other hand" was used inappropriately numerous times. 
  2. The authors need to discuss factors that regulate muscle atrophy more appropriately. They should mention that muscle atrophy develops as a result of an imbalance between protein synthesis and protein degradation. They should cite references related to inhibition of protein synthesis in muscles of ICU patients and that this is coupled with increased protein degradation. They should also mention the four pathways of protein degradation in skeletal muscles (ubiquitin-proteasome, calpains, caspases and autophagy-lysosome pathways). 
  3. The section on diaphragm atrophy during mechanical ventilation is quite weak and confusing. This section needs heavy editing. 
  4. The review would benefit from a diagram showing various fragments of Titin in comparison to full Titin structure. 
  5. How does Titin N fragment analysis compares to other markers of muscle atrophy? 

Author Response

Responses to Reviewer #2:

  1. The manuscript requires extensive editing. Many of the sentences are unclear and the expression "on the other hand" was used inappropriately numerous times.
  2. We have conducted extensive editing, and English editing company (ENAGO) conducted extensive English revision. We revised “on the other hand” as following.

Page 2, Line 46

On the other hand, Another study group, by Nakano et al., reported that urinary titin can be a possible biomarker of muscle atrophy and ICU-AW

Page 4, Line 161

Furthermore, Lindqvist et al. suggested that the positive-end expiratory pressure (PEEP) during ventilation led to the breakdown of the diaphragm titin because the PEEP stretched out the sarcomere of the diaphragm muscle fibers

Page 5, Line 196

Futhermore, O’ Rourke et al. reported that percutaneous electrical phrenic nerve stimula-tion increased diaphragm thickness by 15.1% within 48 hours

Page 5, Line 236

With regard to critical illness neuropathy, Chen et al. investigated the role of titin in neuropathy

Page 6, Line 260

In another study, Nakano et al. investigated 50 consecutive critically ill patients, and found that the medical research council score was lower in the high urinary titin N-fragment group (37.0 [24.0–56.0] vs. 56.0 [51.0–60.0], p = 0.023) [8].

Page 7, Line 294

On the other hand, In COVID-19, muscle weakness lower than 48 was observed in 66% of patients

  1. The authors need to discuss factors that regulate muscle atrophy more appropriately. They should mention that muscle atrophy develops as a result of an imbalance between protein synthesis and protein degradation. They should cite references related to inhibition of protein synthesis in muscles of ICU patients and that this is coupled with increased protein degradation. They should also mention the four pathways of protein degradation in skeletal muscles (ubiquitin-proteasome, calpains, caspases and autophagy-lysosome pathways).
  2. We added the description about protein synthesis and degradation, cited several references, and mentioned four pathways as following.

Page 3, Line 88

Muscle atrophy is caused by an increase in protein degradation or decrease in protein synthesis. Increased protein degradation occurs mainly due to inflammation and immobilization. Calpain, caspase, ubiquitin-proteasome, and autophagy-lysosome pathways have been implicated for this protein degradation pathways [21-23]. De-creased protein synthesis is caused by suppressed insulin-like growth factor-1 and in-activated myogenesis [24].

In critically ill patients, inflammation is an important cause of muscle atrophy [25], and various inflammatory cytokines are associated with muscle atrophy [26]. Immobi-lization is frequently observed during critical illness and causes muscle atrophy [27]. Malnutrition causes a decrease in protein synthesis [28]. In critically ill patients, rec-ommended protein intake is 1.2–2.0 g/kg/day [29], but this level of intake is often not achieved in the ICU [30], resulting in decreased protein synthesis.

  1. The section on diaphragm atrophy during mechanical ventilation is quite weak and confusing. This section needs heavy editing.
  2. We conducted heavy editing as following.

Page 4, Line 148

Diaphragm atrophy is observed in 60% of mechanically ventilated critically ill patients [42], and it is a serious problem because of its association with prolonged mechanical ventilation and prolonged ICU stay [43]. As with limb muscle atrophy, diaphragm atrophy is caused by calpain, caspase, ubiquitin-proteasome, and autophagy-lysosome pathways [44-47]. As reported in limb muscles, inflammation and immobilization are important causes of diaphragm atrophy. Sepsis is a cause of diaphragm atrophy [48], and deep sedation causes immobilization of the diaphragm [49]. Most importantly, ventilator settings strongly influence diaphragm atrophy and subsequent diaphragm dysfunction. Thus, diaphragm dysfunction in such cases is termed as ventilator induced diaphragm dysfunction [50].

Titin plays an important role in the diaphragm contractile force [51], and titin loss has been associated with diaphragm dysfunction in rats [52, 53]. In the diaphragm biopsy of human subjects, Hussain et al. found that prolonged controlled mechanical ventilation decreased titin levels and impaired the diaphragm myofibrillar force [54]. Furthermore, Lindqvist et al. has suggested that the positive-end expiratory pressure (PEEP) during ventilation led to the breakdown of the diaphragm titin because the PEEP stretched out the sarcomere of the diaphragm muscle fibers [55]. Excessive extension may be detrimental to diaphragm titin.

Although titin is associated with diaphragm atrophy and dysfunction, urinary titin N-fragment is not useful to detect diaphragm atrophy. Our previous study investigated the change of diaphragm thickness in 50 critically ill patients using ultrasound. Diaphragm atrophy, defined by >10% decrease of diaphragm thickness, was observed in 32 patients (64%), and the mean diaphragm thickness decreased by 4.9% ± 15.8%, 8.0% ± 16.9%, and 15.4% ± 10.2% on days 3, 5, and 7, respectively. On comparing the diaphragm atrophy and unchanged groups, the levels of urinary titin N-fragment were not higher in the diaphragm atrophy group than those in the unchanged group (147.9 vs. 192.4 pmol/mg in the unchanged vs. atrophy group, p = 0.33) [9]. The urinary titin N-fragment can measure the titin breakdown product in all muscles, and is not specific to the diaphragm. To quantify the diaphragm atrophy, a diaphragm-specific titin kit is necessary. It may be theoretically possible because a cardiac-specific titin kit has been developed in another study [56].

            Interestingly, several studies have reported that increased diaphragm thickness has worsened clinical outcomes as well as diaphragm atrophy [42, 43, 57]. Insufficient ventilatory support leads to excessive respiratory effort in mechanically ventilated patients. This condition increases the diaphragm thickness. Since the increased muscle thickness has worsened outcomes, the hypertrophied diaphragm may not have functional titin to function appropriately. In a previous study on urinary titin N-fragment, there was no significant difference in the cumulative urinary titin N-fragment between the unchanged diaphragm thickness and increased diaphragm thickness groups (147.9 [79.0–257.8] vs. 426.1 [140.8–578.2] pmol/mg Cr in unchanged vs. increased, p = 0.45) [9]. The combination of increased diaphragm thickness and atrophy also did not have a significant difference in terms of the cumulative level of urinary titin N-fragment (147.9 [79.0–257.8] vs. 206.5 [99.3–440.8] pmol/mg Cr in unchanged vs. combination, p = 0.31). The mechanism underlying the increase in diaphragm thickness remains to be elucidated.

            Diaphragm dysfunction is preventable and reversible. Therefore, it is therefore important to maintain spontaneous breathing and avoid excessive ventilatory support during mechanical ventilation, which is called diaphragm protective ventilation [43]. Diaphragm protective ventilation can prevent diaphragm atrophy, compared with lung protective ventilation [58]. Furthermore, O’ Rourke et al. reported that percutaneous electrical phrenic nerve stimulation increased diaphragm thickness by 15.1% within 48 h [59]. Extracorporeal support is also considered to prevent diaphragm injury [60], and in a case report, the early initiation of extracorporeal support prevented diaphragm atrophy, with a relatively suppressed level of urinary titin N-fragment of 24.1–38.4 pmol/mg Cr [61].

  1. The review would benefit from a diagram showing various fragments of Titin in comparison to full Titin structure.
  2. We added the fragments of titin in Fig. 1 and figure legend as following.

In schematic illustration of titin structure (lower right), titin is composed of Z-disk, I-band, A-band, and M-line regions, and the I-band includes Ig and PEVK (Pro-Glu-Val-Lys) domains.

  1. How does Titin N fragment analysis compares to other markers of muscle atrophy?
  2. We compared urinary titin N-fragment with creatinine kinase, BUN/Cr, and urinary Cr as following.

Page 4, Line 139

Unlike other promising biomarkers of muscle atrophy, urinary titin N-fragment is noninvasive and reliable. Creatinine kinase and BUN/Cr are possible biomarkers for muscle atrophy [39], but these biomarkers require blood tests. Furthermore, creatinine kinase is derived from various tissues [40], and BUN/Cr is influenced by various condi-tions including dehydration [39]. Urinary creatinine is also suggested to be a biomarker of muscle atrophy, but it does not consider the kidney function [41]. Thus, urinary titin N-fragment, corrected by urinary creatinine, is a reliable biomarker because it does not depend on kidney function [9, 10].

Reviewer 3 Report

In this review, Nobuto Naknishi discussed and summarized urinary titin N-fragment as a biomarker of muscle atrophy. Titin functions as a molecular spring in sarcomeres. Patients in an intensive care unit (ICU) have been shown the loss of titin, leading to ICU-acquired weakness (ICU-AW). Furthermore, after the ICU discharge, the ICU-AW induces physical dysfunction. The loss of titin can be measured by urinary titin N-fragment which is non-invasive to test ICU-AW.

Major comments)

The author addressed the correlation between the loss of title and immobilization, muscular dystrophy. However, molecular mechanisms of how and why the titin levels are decreased in immobilization and muscular dystrophy are required to explain.

The urinary titin N-fragment reflects the breakdown products of titin in the body. The author’s previous research reported that the urinary titin N-fragment was not higher in the diaphragm atrophy. Explain more detail about the author’s previous research including the period of ICU and the period of mechanical ventilation. And explain how the increased urinary titin N-fragment represents the diaphragm atrophy only, because the patients in ICU also have been shown limb atrophy.

Editing of English is required in many sentences. As a reviewer and reader, I feel that this manuscript is not ready to be submitted. For example,

Line 53, to follow-up -> follow up

Line 89, become -> becomes

Line 122, exhibits -> exhibit

Line 141, 144, has worse -> has worsen

Line 312, Two studies groups -> two study groups

Minor comments)

  • In the introduction (line 38), the author used one review paper as a reference. More research articles are needed.
  • In the introduction (line 39), the word of “acquired” is used repeatedly.
  • In the muscle atrophy part, explain mechanisms of how immobilization reduces titin levels.
  • In lines 141-142, only 2 references are used for several studies. More references are required.

Author Response

Responses to Reviewer #3:

1. The author addressed the correlation between the loss of title and immobilization, muscular dystrophy. However, molecular mechanisms of how and why the titin levels are decreased in immobilization and muscular dystrophy are required to explain.

This is an important point, but the mechanism of titin loss is mostly not revealed. We added the information about the molecular mechanism of titin breakdown as much as possible.

Page 3, Line 130

The molecular mechanism underlying titin breakdown remains unclear. Several pathways, activated by inflammation and immobilization, are involved in the breakdown of titin. Calpain contributes to the cleavage of titin because titin has calpain-binding sites [37]. Lang et al. investigated the ubiquitination of titin in denervated mouse, and found that levels of ubiquitinated titin gradually increased in denervation-induced muscle atrophy [38]. Consistent with this finding, Swist et al. found increased levels of ubiquitinated titin in patients with critical illness [8]. In their study, the markers of autophagy-lysosome pathway were also upregulated. Thus, the autophagy-lysosome pathway may be involved in the breakdown of titin.

2. The urinary titin N-fragment reflects the breakdown products of titin in the body. The author’s previous research reported that the urinary titin N-fragment was not higher in the diaphragm atrophy. Explain more detail about the author’s previous research including the period of ICU and the period of mechanical ventilation. And explain how the increased urinary titin N-fragment represents the diaphragm atrophy only, because the patients in ICU also have been shown limb atrophy.

We added the explanation about our previous research, and explained urinary titin N-fragment cannot distinguish diaphragm atrophy from limb atrophy as following.

Page 4, Line 166

Although titin is associated with diaphragm atrophy and dysfunction, urinary titin N-fragment is not useful to detect diaphragm atrophy. Our previous study investigated the change of diaphragm thickness in 50 critically ill patients using ultrasound. Diaphragm atrophy, defined by >10% decrease of diaphragm thickness, was observed in 32 patients (64%), and the mean diaphragm thickness decreased by 4.9% ± 15.8%, 8.0% ± 16.9%, and 15.4% ± 10.2% on days 3, 5, and 7, respectively. On comparing the diaphragm atrophy and unchanged groups, the levels of urinary titin N-fragment were not higher in the diaphragm atrophy group than those in the unchanged group (147.9 vs. 192.4 pmol/mg in the unchanged vs. atrophy group, p = 0.33) [9]. The urinary titin N-fragment can measure the titin breakdown product in all muscles, and is not specific to the diaphragm. To quantify the diaphragm atrophy, a diaphragm-specific titin kit is necessary. It may be theoretically possible because a cardiac-specific titin kit has been developed in another study [56].

3. Editing of English is required in many sentences. As a reviewer and reader, I feel that this manuscript is not ready to be submitted. For example,

Line 53, to follow-up -> follow up

Line 89, become -> becomes

Line 122, exhibits -> exhibit

Line 141, 144, has worse -> has worsen

Line 312, Two studies groups -> two study groups.

We revised the point, and English editing company (ENAGO) conducted extensive English revision.

4. In the introduction (line 38), the author used one review paper as a reference. More research articles are needed.

We added two more references.

5. In the introduction (line 39), the word of “acquired” is used repeatedly.

We revised the point as following.

Page 1, Line 38

In particular, muscle weakness has been widely recognized as an intensive care unit-acquired weakness (ICU-AW)

6. In the muscle atrophy part, explain mechanisms of how immobilization reduces titin levels.

Immobilization increases urinary titin level, but the molecular mechanism of titin breakdown is mostly unknown. We added the information as much as possible.

Page 3, Line 130

The molecular mechanism underlying titin breakdown remains unclear. Several pathways, activated by inflammation and immobilization, are involved in the breakdown of titin. Calpain contributes to the cleavage of titin because titin has calpain-binding sites [37]. Lang et al. investigated the ubiquitination of titin in denervated mouse, and found that levels of ubiquitinated titin gradually increased in denervation-induced muscle atrophy [38]. Consistent with this finding, Swist et al. found increased levels of ubiquitinated titin in patients with critical illness [8]. In their study, the markers of autophagy-lysosome pathway were also upregulated. Thus, the autophagy-lysosome pathway may be involved in the breakdown of titin.

7. In lines 141-142, only 2 references are used for several studies. More references are required.

We added one more reference.

Round 2

Reviewer 2 Report

No further comments. The authors have edited the manuscript according to my recommendations

Reviewer 3 Report

The authors revised the reviewers' comments properly.